# Platelet Toll-like Receptor 4–Related Innate Immunity Potentially Participates in Transfusion Reactions Independent of ABO Compatibility: An Ex Vivo Study

**DOI:** 10.3390/biomedicines10010029

**Published:** 2021-12-23

**Authors:** Chien-Sung Tsai, Mei-Hua Hu, Yung-Chi Hsu, Go-Shine Huang

**Affiliations:** 1Department of Surgery, Division of Cardiovascular Surgery, Tri-Service General Hospital, Taipei 114, Taiwan; sung1500@mail.ndmctsgh.edu.tw; 2Department and Graduate Institute of Pharmacology, National Defense Medical Center, Taipei 114, Taiwan; 3Institute of Pharmacy, I.M. Sechenov First Moscow State Medical University, Moscow 119991, Russia; 4Department of Pediatrics, Division of Pediatric General Medicine, Chang Gung Memorial Hospital at Linkou, College of Medicine, Chang Gung University, Taoyuan 333, Taiwan; p65952@gmail.com; 5Graduate Institute of Clinical Medical Sciences, College of Medicine, Chang Gung University, Taoyuan 333, Taiwan; 6National Defense Medical Center, Department of Anesthesiology, Tri-Service General Hospital, Taipei 114, Taiwan; X0939778570@gmail.com

**Keywords:** toll-like receptor 4, innate immunity, platelet, blood mixing, transfusion reaction

## Abstract

The role of platelet TLR4 in transfusion reactions remains unclear. This study analyzed platelet TLR4 and certain damage-associated molecular patterns (DAMPs) and evaluated how ABO compatibility affected TLR4 expression after a simulated ex vivo transfusion. A blood bank was the source of donor red blood cells. Blood from patients undergoing cardiac surgery was processed to generate a washed platelet suspension to which the donor blood was added in concentrations 1, 5, and 10% (*v/v*). Blood-mixing experiments were performed on four groups: a 0.9% saline control group (*n* = 31); a matched-blood-type mixing group (group M, *n* = 20); an uncross-matched ABO-specific mixing group (group S, *n* = 20); and an ABO-incompatible blood mixing group (group I, *n* = 20). TLR4 expression in the platelets was determined after blood mixing. We evaluated levels of TLR4-binding DAMPs (HMGB1, S100A8, S100A9, and SAA), lipopolysaccharide-binding protein, and endpoint proteins in the TLR4 signaling pathway. In the M, S, and I groups, 1, 5, and 10% blood mixtures significantly increased TLR4 expression (all *p* < 0.001) in a concentration-dependent manner. Groups M, S, and I were not discovered to have significantly differing TLR4 expression (*p* = 0.148). HMGB1, S100A8, and S100A9 levels were elevated in response to blood mixing, but SAA, lipopolysaccharide-binding protein, TNF-α, IL-1β, and IL-6 levels were not. Blood mixing may elicit innate immune responses by upregulating platelet TLR4 and DAMPs unassociated with ABO compatibility, suggesting that innate immunity through TLR4-mediated signaling may induce transfusion reactions.

## 1. Introduction

Blood transfusions are critical interventions, particularly for patients undergoing hemorrhagic shock. However, despite significant advancements in the safety of blood transfusions, such procedures are still associated with major risks. During blood transfusions, the mechanisms of immune reactions, specifically innate immune reactions, are unclear. Related reactions can result from transfusion with cross-matched blood, uncross-matched type-specific blood, or ABO-incompatible blood [1]. Immune reactions to blood transfusions involve complex interactions among various soluble factors and immune cells, including platelets [1].

Platelets have a major role in thrombosis and hemostasis and are known to have a mediating effect on the body’s innate immune response [2]. They upregulate Toll-like receptor 4 (TLR4), serve as sentinels in the immune system, and are essential for stimulating adaptive immune responses [2,3,4,5,6,7]. We hypothesized that blood transfusion could induce platelet-mediated innate immune reactions caused by the interaction between TLR4 and damage-associated molecular patterns (DAMPs) [8]. Thus, we aim to evaluate the levels of TLR4, lipopolysaccharide (LPS)-binding protein, and certain DAMPs, such as S100 calcium-binding protein A8 (calgranulin A, S100A8, and MRP8), high mobility group box 1 (HMGB1), serum amyloid A (SAA), and S100 calcium-binding protein A9 (S100A9, MRP14, and calgranulin B), in a simulated cross-reaction ex vivo. Furthermore, proinflammatory cytokines downstream of TLR4 signaling, such as proinflammatory cytokines (IL-6, TNF-α, and IL-1β), were evaluated [8]. This study might be valuable since it is the leading study to evaluate the relationship between transfusion and innate immunity through TLR4 bind with DAMPs then test signal transduction.

## 2. Material and Methods

### 2.1. Reagents and Flow Cytometry

The antibodies employed were anti-TLR4-PE antibody (BD Biosciences, Franklin Lakes, NJ, USA), a monoclonal antibody targeting TLR4 expressed on the platelet cell surface; and anti-CD41a-FITC (BD Biosciences), which is a platelet-specific monoclonal antibody that, without needing to be activated, recognizes the platelet GPIIb/IIIa complex. Nonspecific binding was investigated using IgG1 κ-FITC and IgG1 κ-PE antibodies (BD Biosciences). Platelets were stimulated using thrombin (Sigma, St. Louis, MO, USA). In our laboratory, we prepared the pH 7.4 platelet wash buffer (20 mM HEPES, 145 mM NaCl, and 9 mM Na_2_EDTA) and pH 7.4 HEPES-buffered Tyrode’s solution (10 mM HEPES, 1.61 mM KCl, 0.42 mM Na_2_HPO_4_, 11.9 mM NaHCO_3_, 136.89 mM NaCl, 1.05 mM MgCl_2_, and 5.6 mM glucose).

### 2.2. Blood Sampling

Approval for this research was granted by the Tri-Service General Hospital Institutional Review Board (TSGHIRB 1-102-05-014, TSGHIRB 1-107-05-015), and before enrollment, all participants provided written informed consent. Rh-negative patients were excluded from this study. Red blood cells (unwashed) with a 55–60% hematocrit result were obtained from our hospital blood bank and stored at 2–4 °C. We obtained blood samples from an arterial catheter before anesthesia induction in individuals scheduled to undergo cardiac surgery. Each sample was treated with an anticoagulant, which consisted of a 1:9 volume of 3.8% sodium citrate solution. Blood transfusions were simulated ex vivo through blood mixing. Patients were divided into four groups in terms of blood mixing reactions: a 0.9% saline control group, a matched-blood-type mixing group (group M), an uncross-matched ABO-specific mixing group (group S), and an ABO-incompatible mixing group (group I).

The primary outcome measures were TLR4 expression differences among three groups (I, S, and M) and the influence of blood mixing on platelet TLR4 expression. The secondary outcome measures were the influence of blood mixing on the levels of LPS-binding protein and DAMPs, including HMGB1, SAA, S100A8, and S100A9. Furthermore, the levels of molecules downstream of TLR4 signaling were evaluated in each of the three groups (I, S, and M) [8].

### 2.3. Flow Cytometry Analysis of TLR4 Expression

Whole blood underwent centrifugation at 37 °C for 10 min at 200× *g*. We collected the upper phase (plasma) carefully, and we disposed of the interphase (buffy coat chiefly comprising leukocytes and some platelets) and lower phase (red cells). Subsequently, the upper phase was subjected to centrifugation (10 min at 2000× *g*), and buffer was employed to gently resuspend the resultant pellet for washing platelets. For preparing washed platelets, we subjected the suspension to centrifugation (10 min at 2000× *g*) and resuspended the platelets in HEPES-buffered Tyrode’s solution. Subsequently, the suspension was altered so that the final number of platelets was 150,000/μL to 450,000/μL. Thereafter, the washed platelets without or with 0.9% saline added at 1, 5, or 10% (*v/v*) were used as controls. For blood mixing reactions, donor red blood cells were immediately mixed with the washed platelets at 1, 5, or 10% (*v/v*) and incubated at 37 °C for 5 min.

### 2.4. Quantification of TLR4 Expression in Washed Platelets

We investigated platelet TLR4 expression in mixed samples in the presence of thrombin (final concentration: 0.2 U/mL) by incubating the samples at an ambient temperature of 23–26 °C for 5 min. Thrombin is generated during tissue injury, such as that sustained during cardiac surgery [9]. It is a key component of the blood coagulation cascade and a potent platelet stimulator. Thrombin is commonly used in sample preparation protocols for platelet analysis. To quantify the TLR4 expression, sample staining was conducted using a saturating concentration of anti-CD41a-FITC and anti-TLR4-PE monoclonal antibodies, and the samples were incubated for 20 min in a dark environment at 22–26 °C. Subsequently, before they underwent flow cytometry analysis, the samples were fixed using paraformaldehyde (1%) for 30 min at a temperature of 4 °C. PE-labeled IgG1κ and FITC-labeled IgG1κ functioned as background controls. We performed identification of each platelet by using side scattering (granularity features) and anti-CD41a-FITC immunofluorescence on a logarithmic-scaled dot plot. This article presents the results as the mean fluorescence intensity (MFI) of TLR4-PE expression, and for each sample, we gathered reads from 10,000 platelets.

### 2.5. ELISA Analysis

We assessed the levels of LPS-binding protein and of certain DAMPs that may interact with TLR4 after blood mixing. Donor red blood cells were immediately added to whole blood from recipients at 1, 5, or 10% (*v/v*) and incubated for 30 min at 37 °C. The mixed blood then underwent centrifugation (10 min, 37 °C, 100× *g*). We carefully gathered the upper phase (plasma), but we disposed of the interphase (buffy coat chiefly comprising leukocytes and some platelets) and lower phase (red cells). The plasma was analyzed using ELISA kits (in accordance with manufacturer instructions) to determine the concentrations of HMGB1 (Aviva Systems Biology, San Diego, CA, USA), S100A8 (Circulex, MBL, Nagano, Japan), S100A9 (Circulex, MBL), SAA (Abnova Co., Taipei, Taiwan), and LPS-binding protein (Aviva Systems Biology).

### 2.6. Multiplex Immunoassays

The amounts of proinflammatory cytokines IL-6, TNF-α, and IL-1β in the plasma samples were determined using the ProcartaPlex Multiplex Immunoassay (Affymetrix eBioscience, Thermo Fisher Scientific, Waltham, MA, USA) by following the manufacturer’s instructions.

### 2.7. Statistical Analysis

One-way ANOVA was performed for a comparison of the four groups’ demographic variables (control, I, M, and S groups). Furthermore, ANOVA was performed to compare the levels of TLR4, LPS-binding protein, DAMPs (e.g., HMGB1, S100A8, S100A9, and SAA), and TLR4-regulated cytokines (e.g., IL-6, TNF-α, and IL-1β) in each of three groups (M, S, and I), with Scheffé’s post hoc test then performed. Two-way ANOVA was employed to identify differences between three groups (I, M, and S) over various concentrations. All tests were two-sided, and statistical significance was indicated by *p* < 0.05. SPSS (version 20; SPSS, Inc., Chicago, IL, USA) was used for all analyses.

## 3. Results

Demographic characteristics did not differ significantly among the four groups (Table 1). Regarding platelet TLR4 expression, the flow cytometric analysis findings are displayed in Appendix A as fluorescence dot plots showing TLR4 expression in isotype controls (Appendix A ), unstimulated platelets (Appendix A), and platelets stimulated using thrombin not mixed with (Appendix A) and mixed with 10% (Appendix A) donor blood. The histograms for these latter three are overlaid in Appendix A.

### 3.1. Effect of TLR4 Expression after the Addition of 0.9% Saline

We examined the control groups to reveal whether TLR4 expression levels could be determined in blood-mixing experiments by using a washed platelet mixed with 0.9% saline (1, 5, and 10% (*v/v*)). The TLR4 expression levels did not differ significantly following stimulation with thrombin or the 0.9% saline (1, 5, and 10% (*v/v*); *p* = 0.892, Figure 1).

### 3.2. Effect of TLR4 Expression after the Addition of Donor Red Blood Cells

The 1, 5, and 10% (*v/v*) blood mixtures protocols induced a significant increase in TLR4 expression levels in groups I, M, and S (all *p* < 0.001; Figure 2), and the increase was concentration dependent. Compared with the control level, TLR4 expression was increased by 147, 288, and 381% after 1, 5, and 10% blood mixing, respectively (*n* = 60, *p* < 0.001; Figure 2). In group M, TLR4 expression increased by 143, 308, and 399% after 1, 5, and 10% mixing, respectively, compared with the control level (*n* = 20, *p* < 0.001; Figure 2), whereas the increases in group S were 157, 291, and 380% (*n* = 20, *p* < 0.001; Figure 2) and in group I were 142, 265, and 362%, respectively (*n* = 20, *p* < 0.001; Figure 2). However, the three experimental groups’ TLR4 expression levels were not significantly different (*p* = 0.148; Figure 2).

### 3.3. Effect of DAMPs and LPS-Binding Protein in Plasma

HMGB1, S100A8, and S100A9 levels were significantly increased after blood mixing (Figure 3). Compared with the control level, the total HMGB1 levels were significantly increased by 122, 200, and 272% following 1, 5, and 10% mixing, respectively (*p* < 0.001; Figure 3). Total S100A8 levels were 104, 114, and 123% higher (*p* < 0.001; Figure 3) and total S100A9 levels 115, 138, and 150% higher, respectively (*p* = 0.012; Figure 3). However, total SAA levels were not significantly increased in any of the three experimental groups (I, M, and S: *p* = 0.988, 0.588, and 0.999, respectively; Figure 3).

### 3.4. Effect of Proinflammatory Cytokines Downstream of the TLR4 Signaling Pathway in Plasma

Levels of total LPS-binding protein involved in TLR4 signaling (*p* = 0.526) and endpoint proteins, namely total TNF-α, IL-1β, and IL-6 (*p* = 0.998, 0.806, and 0.87, respectively), were not elevated after blood mixing (Figure 4).

## 4. Discussion

This study revealed that after ex vivo blood mixing, TLR4 expression was upregulated in platelets in groups M, S, and I. Besides its important role in thrombosis and hemostasis, platelet TLR4 makes a major contribution to increasing immune and inflammatory responses [2]. This may also be true for platelet TLR4-related immune and inflammatory responses after a transfusion [2]. Platelet TLR4 expression influences innate immunity [5,7,8,10,11], leading to an adaptive immune response [3,4,5,6,7] and inflammation [12], as noted in transfusion reactions. Thus, platelet TLR4 may be a pathophysiological link between innate immunity and transfusion reactions.

TLR4 recognizes DAMPs [13], which are associated with host cell components released after cell damage [5,14,15]. We revealed that levels of certain DAMPs, including HMGB1, S100A8, and S100A9, were elevated after blood mixing, and this finding was corroborated by other reports [16,17,18,19,20] on blood transfusions (Figure 3). For example, first, a transfusion of red blood cells increases vulnerability to lung inflammation through HMGB1 release and induces lung endothelial cell necroptosis [19]. Second, stored human red blood cells contain soluble HMGB1, the levels of which are elevated during storage [16]. Third, salvaged blood analyses identified sustained high levels of certain DAMPs, including S100A8 and S100A9. Transfusion reactions may result from increased levels of TLR4 and certain DAMPs, including HMGB1, S100A8, and S100A9, that bind to TLR4 (Figure 3). The upregulation of TLR4 and certain DAMPs suggests that innate immunity conferred through TLR4-mediated signaling can induce a transfusion reaction.

Upregulated TLR4 expression and elevated levels of some DAMPs (HMGB1, S100A8, and S100A9) were observed in response to blood mixing. However, this action did not involve SAA and LPS-binding protein and did not lead to downstream proinflammatory cytokine (e.g., IL-6, TNF-α, and IL-1β) release after blood mixing, as shown in Figure 4 [8]. These findings may be explained as follows: First, LPS-binding proteins, which are soluble, are synthesized by hepatocytes, present in blood [21], and may not be synthesized in ex vivo studies or detected through ELISA. Second, negative regulators target several TLR4 signaling levels, and various molecules, such as SIGIRR and RP105, impede the initiation of this signaling cascade [8]. Other factors target molecules further downstream of TLR4 signaling through diverse processes [8]. Third, these results may have been obtained because of the short duration of blood mixing (30 min), during which time proinflammatory cytokines were not expressed. Therefore, the interaction between platelet TLR4 and certain DAMPs may not have triggered the proinflammatory cytokine release signaling pathway. Fourth, ex vivo blood mixing could not provide sufficient context in which to determine the full range of platelet responses to transfusion in vivo. Although our results suggest that platelet TLR4 functions as a pathophysiological link between innate immunity and transfusion, why the levels of proinflammatory cytokines downstream of the TLR4 signaling pathway were not increased remains an unanswered question.

Groups M, S, and I did not differ significantly in terms of TLR4 level. The major difference between them was the responses of antibodies recognizing transfusion antigens [22]. In most blood transfusions for humans, red blood cell membranes containing components of the ABO system are crucial features. These antibodies are implicated in the severest transfusion reactions. However, platelets do not have ABO systems [22]. Additionally, surface antigens on red blood cells include Duffy, Kell, Kidd, MNS, and P systems, which are not present on platelets [22]. We revealed no intergroup differences in platelet TLR4 expression, implying that reactions between antigens and antibodies had no role in TLR4 expression induction.

TLR4 ligands recognize DAMPs and pathogen-associated molecular patterns (PAMPs) [5,14,15,23], such as circulating LPS (endotoxin). PAMPs are associated with microbial pathogens [24]. PAMP signaling is unlikely to participate proinflammatory cytokine release signaling pathway for the following two reasons: First, during blood mixing, PAMPs such as circulating LPS from microbial pathogens were not considered because no patients had a pathogenic infection. Second, after blood mixing, the levels of LPS-binding protein, which binds to PAMPs but not DAMPs, were not increased, as shown in Figure 4 [14,15]. Therefore, blood mixing may initiate platelet TLR4 expression, which could trigger an innate immune system response that is likely independent of PAMPs.

The rationale for selecting patients undergoing cardiac surgery in this study was as follows: First, blood transfusion is routinely required during cardiac surgery, and preparation for such surgery routinely requires cross-match testing of blood at least 1 day before the surgery. Therefore, cross-matched red blood cells were available, and they were sent to the operation room from the blood bank before anesthesia, enabling us to obtain donor cells. The cells were also used for heart–lung machine priming or stored until needed for transfusion. Second, patients undergoing cardiac surgery require arterial catheterization for aggressive hemodynamic monitoring. The arterial catheterization approach used for recipient blood sampling before anesthesia and skin incision prevents contamination from anesthetics and tissue damage. The anesthetics and tissue damage may be a confounding factor affecting platelet function and activation (Appendix A). Red blood cells from groups S and I were also obtained from the blood bank for other major surgeries at the study hospital.

Our blood mixing procedure is clinically relevant. We mixed, ex vivo, the donor and recipient blood; this method is similar to the cross-matching approach. Before a blood transfusion, cross-matching is applied to reveal the compatibility of a donor’s blood with that of the recipient. The participants of this study had a mean body weight of 66.8 kg; therefore, their total blood volume was around 4620 mL (approximately 7% of body weight). Our 1–10% blood mixtures thus corresponded to volumes of around 46.8–467.6 mL, respectively, which are commonly employed when administering transfusions.

This study had two limitations. First, the baseline levels of HMGB1, S100A8, S100A9, SAA, LPS-binding protein, IL-6, TNF-α, and IL-1β were undetected in the donor red blood cells. However, these cells were added to the recipient whole blood at various volume concentrations (1, 5, or 10% (*v/v*)). The donor red blood cells had 55–60% hematocrit and these percentages were standardized. Therefore, they contained less plasma than the recipient whole blood did. Therefore, the plasma volume of our donor red blood cells was substantially smaller than that of our recipient whole blood. Second, we primarily used an ex vivo model because we included a group I (ABO-incompatible blood mixing) protocol, which could be harmful to patients if conducted in vivo. Therefore, in vivo host innate immune responses were not evaluated. Platelets have direct involvement in immune defense, and they assist with and regulate various activities of innate immune cells. Furthermore, platelets modulate immune cell function by attaching themselves to immune cells or releasing nucleosides, platelet-derived microvesicles, mitochondrial DNA, growth factors, lipid mediators, chemokines, and cytokines. Platelets and their associated releasates exert diverse effects on migration, microbicidal activity, differentiation, phagocytosis, clearance of pathogens, cytokine responses of innate immune cells, and extracellular trap formation [24].

## 5. Conclusions

Platelet TLR4 has actions at the crossroads of innate immune responses and thrombosis. We revealed that allogeneic blood mixing may modulate innate immune responses by upregulating platelet TLR4 and DAMPs, including HMGB1, S100A8, and S100A9, which may bind to TLR4, suggesting that platelet TLR4 links between transfusion and innate immunity in blood-mixing reactions. Given the numerous platelets that circulate in the blood, potential interactions between platelet TLR4 and DAMPs and an innate immune response is induced, leading to transfusion reactions, are possible. However, because TLR4 downstream proinflammatory cytokines were undetected, it remains unclear whether TLR4 signaling leads to a transfusion reaction under in vivo conditions. TLR4 expression upregulation was not associated with ABO compatibility. These findings indicate that TLR4 contributes to transfusion reactions that are unrelated to antibodies against red blood cell antigens. Whether platelet TLR4 is a novel therapeutic and prophylactic target in transfusion reactions or a new target for modulating innate immunity requires further study.

## Figures and Tables

**Figure 1 biomedicines-10-00029-f001:**
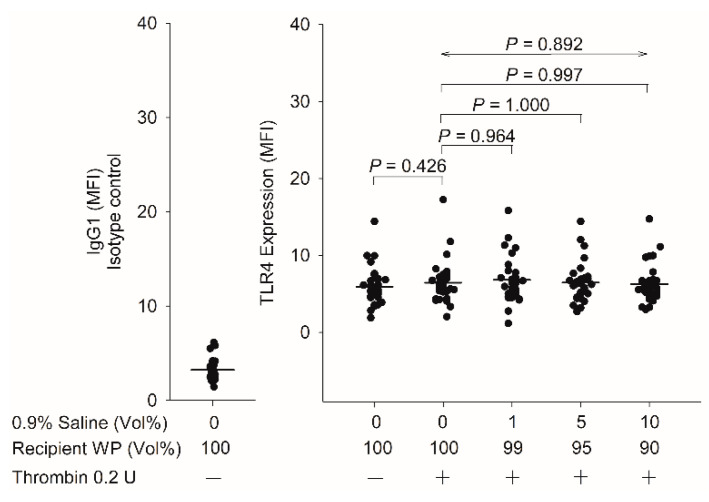
Effect of TLR4 expression after 0.9% saline was mixed with washed platelets (group M, *n* = 10; group S, *n* = 10; and group I, *n* = 11). WP, washed platelets; vol, volume; MFI, mean fluorescence intensity. Data are presented as mean ± SD.

**Figure 2 biomedicines-10-00029-f002:**
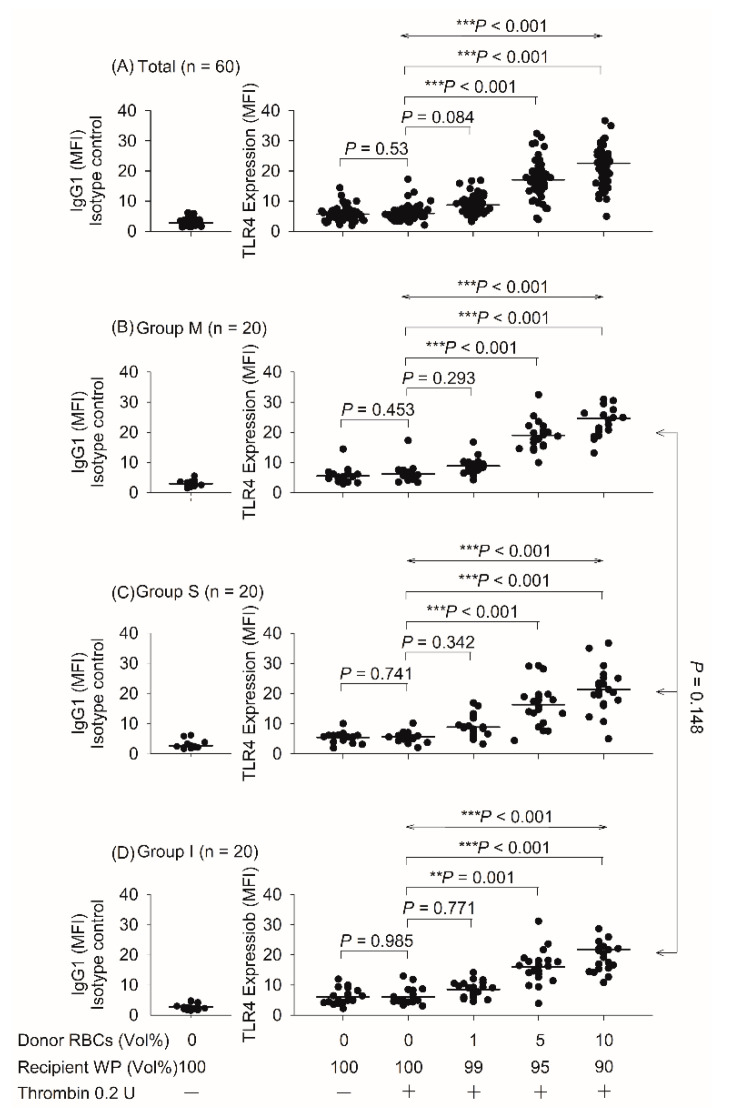
Effect of TLR4 expression on groups M, S, and I after the mixing of donor red blood cells and washed platelets. MFI, mean fluorescence intensity; RBCs, red blood cells; vol, volume; WP, washed platelets. Data are expressed as mean ± SD. ** *p* < 0.01, *** *p* < 0.001.

**Figure 3 biomedicines-10-00029-f003:**
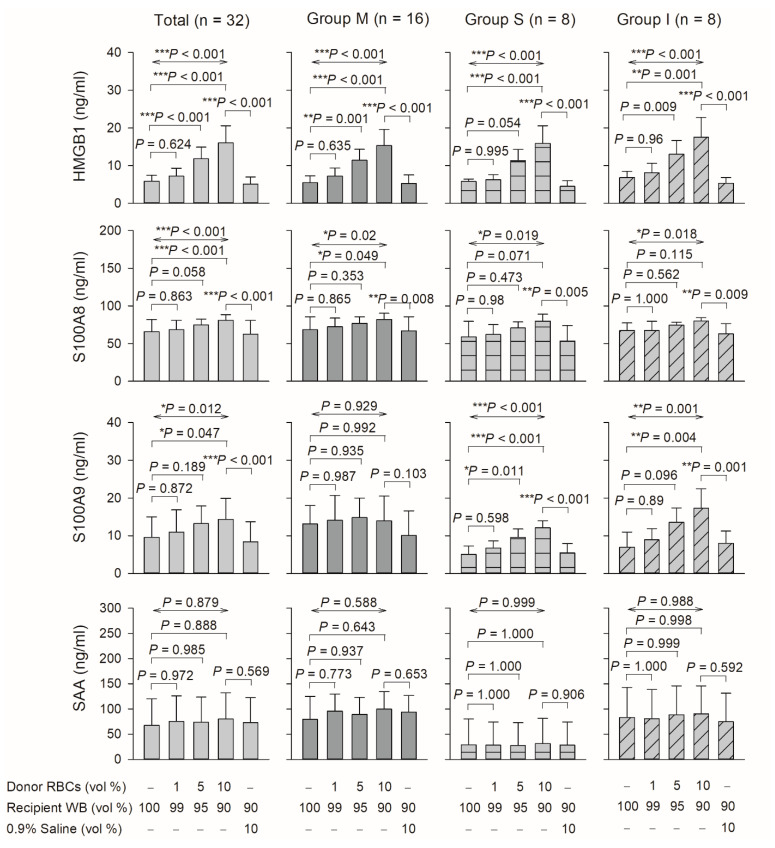
Effect of DAMPs (HMGB1, S100A8, S100A9, and SAA) in the plasma prepared from mixing donor red blood cells and recipient whole blood on total blood mixing and groups M, S, and I. HMGB1, high mobility group box-1; SAA, serum amyloid A; RBCs, red blood cells; vol, volume; WP, washed platelets. Data are expressed as mean ± SD. * *p* < 0.05, ** *p* < 0.01, *** *p* < 0.001.

**Figure 4 biomedicines-10-00029-f004:**
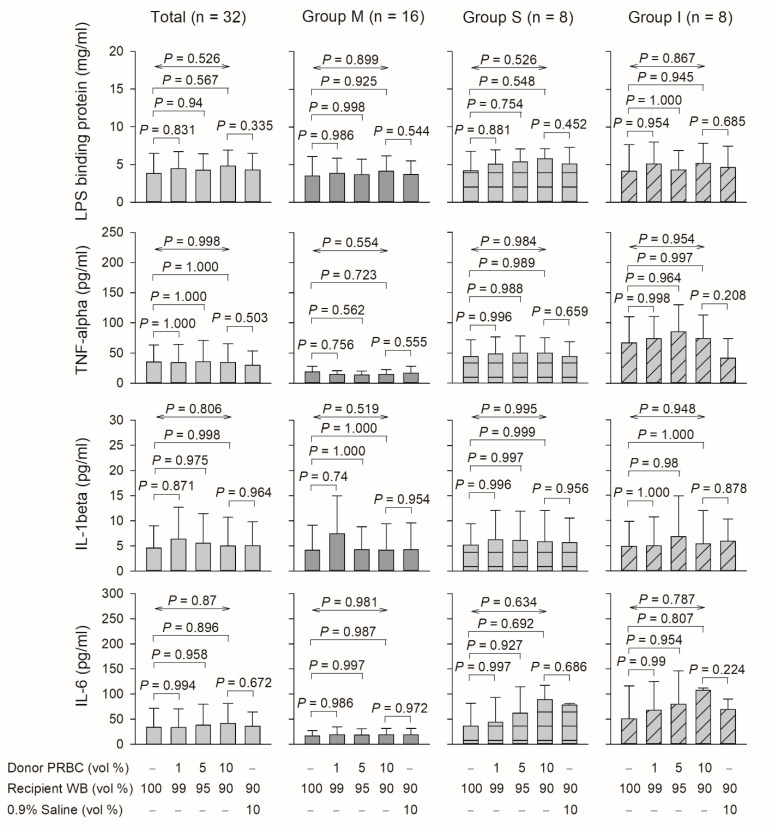
Effect of LPS-binding protein and proinflammatory cytokines in plasma prepared from mixing donor red blood cells and recipient whole blood on groups M, S, and I. RBCs, red blood cells; vol, volume; WP, washed platelets. Data are expressed as mean ± SD.

**Table 1 biomedicines-10-00029-t001:** Demographic characteristics of recipients in 0.9% saline control and groups M, S, and I.

Groups	0.9% Saline Control	M	S	I	*p* Value of Four Groups
Number of cases	31	20	20	20	
Age (years)	60.4 ± 12.0	61.2 ± 10.7	60.1 ± 11.7	65.7 ± 13.7	0.424
Height (cm)	165.0 ± 7.9	165.1 ± 8.2	165.1 ± 7.3	165.8 ± 8.2	0.991
Weight (kg)	67.7 ± 11.9	69.1 ± 13.3	66.1 ± 10.8	65.3 ± 9.2	0.724
Body mass index (kg/m^2^)	24.9 ± 4.2	25.2 ± 4.1	24.2 ± 3.5	23.8 ± 3.4	0.634
Women/Men	12/19	4/16	6/14	7/13	0.497

Data are presented as the mean ± SD. M, matched blood type mixture; group S, uncross-matched ABO group-specific mixture; I, ABO incompatibility blood mixture.

## Data Availability

http://www.chictr.org.cn/showproj.aspx?proj=123019. Accessed date: 22 November 2021.

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
