# Peer review of "Platelet Toll-like Receptor 4–Related Innate Immunity Potentially Participates in Transfusion Reactions Independent of ABO Compatibility: An Ex Vivo Study"

_biomedicines, 2021, doi:10.3390/biomedicines10010029_

Round 1
Reviewer 1 Report
In the article by Chien-Sung Tsai et al., was evaluated how ABO
compatibility affected TLR4 expression after a simulated ex vivo transfusion. This is an interesting article that needs some minor fixes:
- It is necessary to expand the introduction
- Figures of materials and methods could be as supplementary material.
- Before the assays, was platelet function evaluated? (e.g. aggregation, activation ...)
Author Response
Response to Reviewer 1 Comments
Reply to the Review Report (Reviewer 1)
Comments and Suggestions for Authors
In the article by Chien-Sung Tsai et al., was evaluated how ABO
compatibility affected TLR4 expression after a simulated ex vivo transfusion. This is an interesting article that needs some minor fixes:
- It is necessary to expand the introduction
- Figures of materials and methods could be as supplementary material.
- Before the assays, was platelet function evaluated? (e.g. aggregation, activation ...)
Point 1: It is necessary to expand the introduction
Response 1: Thank you very much for your comment. This section was revised including the context for the study, clearly state the research objective (aim), and establish the significance of the study. Please see the “Track Changes” in line 50 and lines 55-58.
Point 2: Figures of materials and methods could be as supplementary material
Response 2: Thank you very much for your comment. We move figure 1 and figure 2 with figure legends into the new created “Supplementary files”. The Label of (Fig. 1) was change to (Supplemental Figure S1); The Label of (Fig. 2) was change to (Supplemental Figure S2) in the main text. Therefore, the order of appearance of the figures were also numbered two in advance, for example: figure 6 is changed to figure 4 in the main text.
Point 3: Before the assays, was platelet function evaluated? (e.g. aggregation, activation ...)
Response 3: Thank you very much for your comment. We didn’t evaluate platelet function before the assays such as aggregation and activation. First of all, we did not expect to detect platelet aggregation and activation in this study. Second, there are too many test tubes and items that must be tested at the same time in this experiment, we can’t do aggregation and activation tests at the same time.

Reviewer 2 Report
The study seems well-conducted and I did not identify any major flaws.
Some suggestions for revision are listed below:
- Please move the figures from the methods section to the results.
- Improve the clarity of the figures.
- Add newer references to the discussion, introduction etc. The references dating from 2017 onwards are too few.
Author Response
Response to Reviewer 2 Comments
Author's Reply to the Review Report (Reviewer 2)
Comments and Suggestions for Authors
The study seems well-conducted and I did not identify any major flaws.
Some suggestions for revision are listed below:
- Please move the figures from the methods section to the results.
- Improve the clarity of the figures.
- Add newer references to the discussion, introduction etc. The references dating from 2017 onwards are too few.
Point 1: Please move the figures from the methods section to the results.
Response 1: Thank you very much for your comment. The figures display in methods were laid out by editors of Biomedicines. I think the office editors was every excellent and professional that will correct the lay out if this manuscript was accepted. I also will remind them in the resubmission covering letter. We move figure 1 and figure 2 with figure legends into a new created “Supplementary files”. In the main text, the Label of (Fig. 1) was change to (Supplemental Figure S1); The Label of (Fig. 1) was change to (Supplemental Figure S1) according to reviewer 1 suggestion. Therefore, the order of appearance of the figures were also numbered two in advance, for example: figure 6 is changed to figure 4 in the main text.
Point 2: Improve the clarity of the figures. We Improve the resolution of all pictures to clarity of the figures
Response 2: Thank you very much for your comment. According to the “Instructions for Authors Biomedicines”. File for Figures and Schemes must be provided during submission in a single zip archive and at a sufficiently high resolution (minimum 1000 pixels width/height, or a resolution of 300 dpi or higher). Common formats are accepted, however, TIFF, JPEG, EPS and PDF are preferred.
Point 3: Add newer references to the discussion, introduction etc. The references dating from 2017 onwards are too few.
Response 3: Thank you very much for your comment. The references we choice was most valid to reflect the key work that led to the question of my paper. However, to select the most recent studies was also very impotent. Some references were revised newer and also fit the principle of valid to reflect the key work that led to the question of my paper. After revision, four references before 2017,five in 2017 and 15 references after 2017. Thank you for your practical suggestion. The revised references were marked up by using the “Track Changes” function, such that changes can be easily viewed by the editors and reviewers.
